# Pediatric Brain Tumours: Lessons from the Immune Microenvironment

Betty Yao [1], Alberto Delaidelli [1,2], Hannes Vogel [3] and Poul H. Sorensen [1,2,*]

1   Department of Molecular Oncology, British Columbia Cancer Research Centre, Vancouver, BC V5Z 1L3, Canada; byyao@student.ubc.ca (B.Y.)
2   Department of Pathology and Laboratory Medicine, University of British Columbia, Vancouver, BC V6T 1Z3, Canada
3   Department of Pathology, School of Medicine, Stanford University, Stanford, CA 94305, USA
*   Correspondence: psor@mail.ubc.ca

**Abstract:** In spite of recent advances in tumour molecular subtyping, pediatric brain tumours (PBTs) remain the leading cause of cancer-related deaths in children. While some PBTs are treatable with favourable outcomes, recurrent and metastatic disease for certain types of PBTs remains challenging and is often fatal. Tumour immunotherapy has emerged as a hopeful avenue for the treatment of childhood tumours, and recent immunotherapy efforts have been directed towards PBTs. This strategy has the potential to combat otherwise incurable PBTs, while minimizing off-target effects and long-term sequelae. As the infiltration and activation states of immune cells, including tumour-infiltrating lymphocytes and tumour-associated macrophages, are key to shaping responses towards immunotherapy, this review explores the immune landscape of the developing brain and discusses the tumour immune microenvironments of common PBTs, with hopes of conferring insights that may inform future treatment design.

**Keywords:** tumour immune microenvironment; pediatric brain tumour; immunotherapy; medulloblastoma; pediatric high-grade glioma; tumour-infiltrating lymphocytes; tumour-infiltrating macrophages





## 1. Introduction

Improvement of pediatric brain tumour (PBT)-related mortality rates have seen limited progress in the past few decades. Among the most common neoplasms that affect pediatric populations, improvements in survival, supportive care, and management of long-term sequelae have disproportionately favoured blood cancers [1]. Since 2011, pediatric brain tumours have surpassed leukemias as the leading cause of cancer-related deaths in patients aged between 1–19 years old [1]. Challenges to the development of effective therapies against solid tumours play a significant role in such a discrepancy. Solid tumours have highly heterogenous cell populations comprised of tumour cells and physiological tissue alike that may support tumour initiation, growth, and further disease progression [2].

Efforts to study the PBT immune microenvironment are often guided by current insights learned from adult brain tumours. Reliance on pre-existing paradigms have their share of merits and limitations; while there is significant intersection in their current standards of care, adult and pediatric brain tumours are each a unique collection of diseases. A robust understanding of the pediatric brain tumour immune microenvironment (TIME) and its underpinnings in both malignant transformation and the development of therapeutic resistance is instrumental for improving the effectiveness of both current and up-and-coming therapies.

This review will first discuss our limited knowledge of immune physiology within the pediatric central nervous system (CNS). Next, it will highlight the diversity of TIMEs present in PBTs, and how they have adapted mechanisms to escape immune surveillance

or even co-opt the immune system to drive tumour development, with a focus on pediatric medulloblastoma and high-grade glioma.

## 2. Immune Surveillance and Trafficking in the Developing Central Nervous System

The developing brain, in conjunction with the immature immune system, harbors profound and appreciable physiological differences from the adult CNS that must be accounted for in disease models. Immune signalling itself is indispensable for normal CNS development [3]. It is no surprise, then, that the immune system can be exploited in disease processes, including the oncogenesis and growth of PBTs. Even in its "immature" state during perinatal and childhood development, the brain enjoys particular benefits from the immune system that pediatric brain tumours are able to accroach [4].

The notion of an "immune-privileged" brain sequestered from the body's immune system, while once a popular paradigm to describe the unique brain environment, has since been disproven: the brain, while unique from other sites in the body, is an immunologically active organ [5]. It is more accurate to consider the brain a highly "immune-specialized" compartment [6] that maintains a "privileged dialogue" with the immune system [7]. While the blood–brain barrier (BBB) restricts circulatory trafficking into the CNS, lymphocytes and other immune cells are present even in the absence of infection [8]. Leukocytes from peripheral blood can access the CNS through several means: limited direct extravasation through the postcapillary venules [8], extravasation into the cerebrospinal fluid (CSF), and subsequent egress into the cervical [9] or meningeal lymph nodes [10,11]. In the context of PBTs themselves, it is also noteworthy that the pediatric BBB does not share the same qualities as its adult counterpart; while the younger, functional BBB still readily allows plasma-derived proteins regulated, transcytosis-mediated access to the brain parenchyma by brain endothelial cells (BECs), the aged, "leaky" BBB is more permissive to the unregulated entry of potentially neurotoxic proteins from peripheral blood [12]. Greater degrees of nonspecific protein transport across the BBB are attributed to an age-associated reduction of BEC pericyte coverage [12,13] and transcriptional changes induced by age-related, blood-borne signals [14]. Glioblastoma, the most common and lethal adult brain tumour [15]—and consequently, whose body of literature comprises much of our knowledge on CNS malignancies overall—emerges under physiological conditions that are inherently more susceptible to inflammation than the childhood CNS that would later shape the TIME. For reader interest, a more detailed review of the age-related changes to brain vasculature is available [16].

The unique immune status of the CNS is also owed, in part, due to its nearby lymphopoietic and myelopoietic niches at the skull and vertebral bone marrow without first being trafficked through systemic blood flow. Locally-sourced myeloid cells, including macrophages and granulocytes [7,17], and B lymphocytes can enter the brain parenchyma through the meningeal lymphatic vessels, even in the absence of infection and inflammation. This privileged dialogue is mediated by the CSF and meninges itself; CSF-derived factors maintain homeostatic signalling with the skull bone marrow and mould the hematopoietic niche [18], including downregulating genes involved in proliferation and the generation of reactive oxygen species. Importantly, the dural sinuses can also become a site of immune surveillance; tumour-derived antigens that enter the CSF may accumulate in the dural sinus stroma, where they are then taken up by resident antigen-presenting cells [19]. The sinus stroma also strongly expresses several chemokine ligands, including CXCL12, which induces CXCR4-mediated T cell chemotaxis. This only recently-characterized route for lymphocyte entry to the brain likely contributes to why intracerebroventricular administration of chimeric antigen receptor (CAR) T cells into the CSF has greater therapeutic efficacy than intravenous infusions into peripheral blood [20,21].

A range of cell signalling components and immune cells themselves also adopt distinct functions within the CNS compared to their traditional functions best studied in non-nervous tissue [22]. While best known for their role in endogenous antigen presentation, allowing cells to distinguish themselves as "self" to the immune system [23], major

histocompatibility complex (MHC) class I receptors are widely expressed in neuronal tissue and engage transmembrane proteins uninvolved in the innate or adaptive immune systems. MHC class I molecules are especially expressed in the perinatal brain [24] as critical participants in normal neurodevelopment [25], including synaptic activity-dependent modifications to neuronal structure [26], and are generally present in greater levels compared to the adult CNS [27]. A range of cell types within the brain express MHC class I molecules, including the surfaces of neural axons and dendrites [26]. MHC class I expression in oligodendrocytes and astrocytes appears to be inducible by interferon-$\gamma$ (IFN-$\gamma$), and is otherwise not constitutive [28]. MHC Class I receptor-expressing CNS tissue is therefore potentially immunogenic and subject to immune surveillance, as well as being capable of interactions with effector immune cells such as natural killer (NK) cells and cytotoxic T lymphocytes (CTLs).

Other components of the innate immune system also emerge early during development: while it is unclear whether they emerge by differentiation within embryonic tissue or colonize nervous tissue from circulating blood, macrophages are active in the brain from as early as 8 weeks post-gestation, where they begin producing cytokines and may initiate toll-like receptor-mediated inflammatory reactions at birth [5]. Microglia, the resident immune sentinels of the brain, are detectable in the fetal neuroaxis from the 4th week of gestation [29]. They act as the primary mediators of neuroinflammation and serve as the main bridge between innate and adaptive immunity in the brain [30]. MHC Class II$^+$ microglia begin to populate neural tissue 18 to 24 weeks post-gestation [29]. Aside from their role in immunosurveillance, they are also key players in neurodevelopment—in addition to serving as a line of defense against infection and disease, they both induce apoptosis in selected neural progenitor cells while favouring the survival and proliferation of others [31]. Over the course of early CNS development, microglial populations make significant adjustments to the final cellular makeup of the brain parenchyma via the secretion of cytokines including but not limited to interleukin-6 (IL-6), tumour necrosis factor-$\alpha$ (TNF-$\alpha$), and IFN-$\gamma$ [32].

Lastly, while largely naïve, the perinatal CNS immune system is capable of mounting an adaptive immune response. Both T and B cells are active components of pediatric brain physiology. The young immune system, however, does not yet carry ample immunological memory and is unable to carry out robust T cell-mediated immune responses [33]. Despite a relatively uneducated immune system, $\gamma\delta$ T cells are still available to provide some degree of immune protection early in life as they can defend the host independently of MHC-dependent pathogen recognition [34,35], and have been found in the neural tissue of injured pediatric brains. The $\gamma\delta$ T cell subset populates the brain early during development to respond against injury and other brain pathologies, including cancer [36].

The knowledge of the immune system within the pediatric brain, while gradually growing more refined, is largely limited to post-mortem studies meant to better understand CNS pathologies rather than normal physiology. Efforts to unearth the dynamic interactions between both resident and infiltrating immune cells within the perinatal CNS, such as via the use of live imaging, are often limited to animal models or in vitro research. Human studies, meanwhile, can only provide "snapshots" into discrete developmental timepoints.

## 3. Immunoediting and Immunomodulation in Pediatric Brain Tumours

Malignant tumours, in part, emerge due to a lapse in effective immunosurveillance [4]—if the pediatric brain is indeed immunologically active, tumour cells capable of evading or withstanding the antitumour activity of the immune system will continue to proliferate as more immunogenic cancer cells are eliminated. Immunologic "sculpting" confers the malignant tumour's ability to both indefinitely escape the host-protecting activity of the immune system and to appropriate the immune system components that have now localized to the tumour site to potentiate its own growth [37].

The aforementioned process is often termed "immunoediting," and can be broken down into three broad phases. During the first elimination phase both arms of the immune

system effectively clear potentially malignant cells [4]. During the equilibrium phase, tumour cell growth and proliferation, while persistent, is sufficiently curbed by the immune system in such a way that there is no net tumour growth. The elimination and equilibrium phases ultimately select less immunogenic cells capable of the final phase of immunoediting—immune escape [38]. During this phase, undetected by the host immune system, the tumour cells are free to proliferate, and clinical signs of disease begin to emerge [39]. Our current models of cancer immunoediting emerge from a series of landmark studies, which have been previously described in excellent detail by several reviews [40–42].

Pediatric cancers, including PBTs, have proclivities toward immune escape. Tumour mutational burden (TMB) is generally low, with PBTs most often sporting less than 2–10 mutations per mega-base (Mb) of genomic DNA [43,44]—markedly less than their adult counterparts. This has been postulated as a reason why it is difficult to find effective immunotherapy approaches for PBTs: there are fewer antigens specific to the tumour itself to target, both by the immune system over the course of immunoediting and while designing immune cell-based therapies that will be efficient in tumour clearance [45].

The TIME also adds an important, yet elusive dimension to overall disease formation and progression. Broadly, PBTs are often considered "immune-cold"; in many cases, they are "defectively primed" towards host-protective immune cell activity (Figure 1), characterized by a relatively low expression of immunogenic markers, the lack of proinflammatory or cytotoxic immune infiltrate or corresponding cytokines, and the presence of regulatory immune cells, including myeloid-derived suppressor cells (MDSCs) and T regulatory ($T_{Reg}$) cells [44,46,47], although a minority of PBTs have greater degrees of inflammation and immunogenicity [48]. The consequences of this immune system downregulation also appear to have systemic effects: lymphopenia, as measured by both absolute cell counts, and high preoperative neutrophil-to-lymphocyte ratios present in peripheral blood, are prognostic of both unfavourable progression-free and overall survival [49,50]. While patients with higher grade tumours such as pediatric medulloblastoma (MBL) have similar neutrophil counts compared to patients with lower-grade pilocytic astrocytoma, absolute blood lymphocyte counts are markedly reduced. Additionally, half of all high tumour grade patients meet the criteria for lymphopenia, alluding to some form of systemic tumour-induced immunosuppression [51,52].

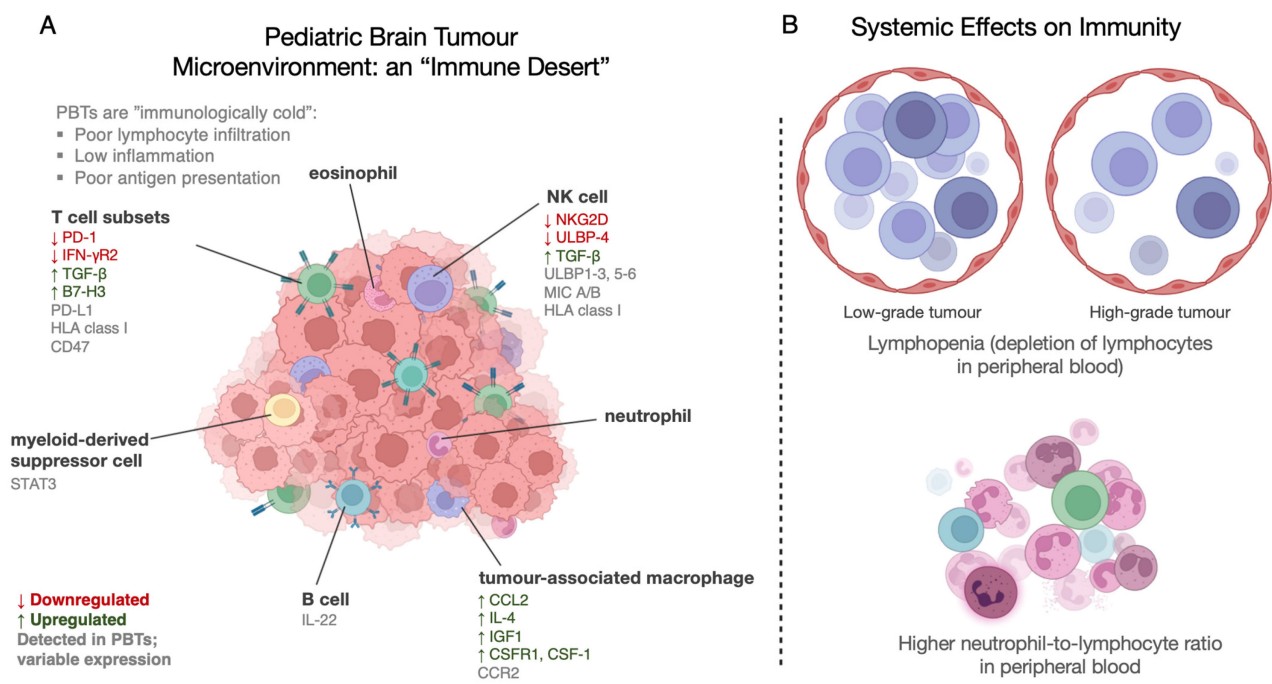

**Figure 1.** Sweeping overview of the host immune system's interactions with PBTs. (**A**) Genes listed have been implicated in tumour cell–immune cell interactions in medulloblastoma and pediatric high-grade glioma. Certain immune checkpoint molecules, chemokines, cytokines, stress ligands characterize the TIME. Genes in red are generally downregulated; upregulated genes are in green; uncoloured genes vary greatly with respect to expression among different tumour types. CD47, cluster of differentiation 47 [53,54]; STAT3, signal transducer and activator of transcription 3 [46]; NKG2D, natural killer group 2D [55]; ULBP-4, UL16 binding protein 4 [55]; ULBP1-3/ULBP5-6, UL16 binding protein 1-3/5-6 [55]; MIC A/B, MHC class I polypeptide related sequence A/B [55]; TGF-β, transforming growth factor beta [55,56]; IL-4, interleukin-4 [57]; IGF1, insulin-like growth factor 1 [57]; HLA class I, human leukocyte antigen class I [56,58]; PD-1, programmed cell death protein 1 [59]; IFN-γR2, interferon-gamma receptor 2 [60]; CSFR1, colony stimulating factor 1 receptor [61]; CSF-1, colony stimulating factor 1 [61]; CCL2, chemokine (C-C motif) ligand 2 [56,62,63]; CCR2, C-C chemokine receptor type 2 [56,62,63]; IL-22, interleukin-22 [64]; B7-H3, B7 Homolog 3 [56,65,66]. (**B**) Systemic alternations to the immune system induced by malignant PBTs.

With immunotherapy becoming a mainstay in our strategies against pediatric cancers, we must learn how to carefully navigate different immune microenvironments. Combating solid tumours propounds an array of challenges, including difficulties in administering treatment and limited surface target availability [67]. Beyond that, we must also take care to properly understand the TIME. To illustrate this, we can perhaps consider the following questions: will a more inflammatory phenotype be conducive to certain immunotherapeutic approaches? If a certain lymphocyte population is present within the TIME, how should a treatment program modify itself accordingly? Immune checkpoint inhibition (ICI) therapies, for example, are administered as a means to potentiate the body's own antitumour response via effector lymphocytes [68]. Accordingly, the most promising responses to ICI are seen in tumours with greater IFN-γ secretion and cytotoxic CD8+ T cell populations [69]. If most PBTs have low lymphocyte numbers to begin with, a possible next step may be to ask what causes such poor infiltration to begin with. A recent immunogenomic study appears to suggest that the nature of the TIME is more accurately predicted by which specific oncogenic pathways have been disrupted in a particular cancer type, rather than a lower TMB. While predicted MHC presentation of strong-binding peptides appears to coincide with greater TMB in hypermutated pediatric high-grade gliomas, there is otherwise no relationship between TMB and the nature of the immune microenvironment in PBTs [70].

The aforementioned study supports the notion that the activation of certain oncogenes can mediate T cell exclusion [71].

A robust understanding of how the tumour interacts with the immune system—and possible therapy-induced perturbations—form the bedrock to treatment design. The remainder of this review thereby aims to outline our current state of knowledge regarding the PBT TIME. Whether an aspect of the TIME supports or impedes disease progression or treatment resistance becomes of interest for therapies. In the context of generally low immunogenicity in PBTs, this could mean identifying components whose aspects of the TIME are contributing to tolerance against the tumour, and developing therapeutic regimens that can overpower these mechanisms to either support the patient's own immune responses or improve the effectiveness of cell-based immunotherapies, such as CAR T cell therapy [72].

### 3.1. Medulloblastoma

Medulloblastomas (MBLs) are embryonal tumours whose four molecular groups comprise some of the most common forms of pediatric brain cancer. While MBLs are probably one of the most extensively studied PBT entities, treatment modalities remain largely unevolved. Radiation therapy, often provided following maximal surgical resection, while effective, produces damaging side effects for the still-developing brain [73,74]. Despite improved patient prognoses for localized MBL in recent decades [75], the vast percentage of patients who go on to develop disseminated disease—with metastases most often occurring in the leptomeningeal space—have unacceptably poor survival outcomes and make up a disproportionate percentage of fatalities. Even children who survive long-term almost uniformly suffer life-altering neurological deficits that stymie independence and cognition as the patient ages [76,77].

Broadly, MBL TIMEs tend to fall into two major categories: they are either immune "neutral," with TIMEs that are considered cold microenvironments with respect to adult tumours, or immune "excluded," where there is an even lower degree of immune system infiltrate in the tumour microenvironment when compared among all pediatric nervous system tumours [70]. The distinct pathogenesis and outcomes of the wingless-activated (WNT) and sonic hedgehog-activated (SHH) Group 3 and Group 4 tumours is reflected by their unique immune microenvironments [78]. Accordingly, the molecular group can be predictive of the composition of the TIME; 83% of SHH MB tumours feature higher proportions of B cells and CD8$^+$ T cells, while Group 3 and Group 4 tumours largely comprise low amounts of infiltrating CD8$^+$ T cells, and otherwise have no significant skew towards a particular immune cell type. Immunophenotyping—through the analysis of a large cohort of tumours via methylCIBERSORT—reveals the predictive capacity of immune cell infiltrate population towards survival outcomes [47]. In SHH MBL, for example, greater $T_{Reg}$ cell populations lead to worse progression-free survival. Group 3 MBL, meanwhile, has a trickier relationship with $T_{Reg}$ cell infiltration: tumours with both high and low proportions of $T_{Reg}$ cells exist within higher- and lower-risk tumours [47].

Our current molecular delineations of MBL disease groups are useful and informative toward disease management and stand to benefit greatly from immuno-profiling (Table 1). Adult brain tumours, too, see correlations in particular immunophenotypes, pathology, and outcomes [79]. Characterization of infiltrating immune cell populations in MBL illustrates both the pressing need to make available more specific, precise therapies to tackle separate risk groups, and acts as a possible approach to further refine patient stratification.

**Table 1.** Immune signalling components reported in MBL. CD47, cluster of differentiation 47 [53,54]; STAT3, signal transducer and activator of transcription 3 [46]; TGF-β, transforming growth factor beta [55]; ULBP-4, UL16 binding protein 4 [80]; IGF1, insulin-like growth factor 1 [57]; ERBB4, receptor tyrosine-protein kinase erbB-4 [81]; HLA class I, human leukocyte antigen class I [58]; PD-L1, programmed death ligand 1 [59]; IFN-γR2, interferon-gamma receptor 2 [60]; CSFR1, colony stimulating factor 1 receptor [61]; CCR2, C-C chemokine receptor type 2 [62]; IL-22, interleukin-22 [64]; B7-H3, B7 Homolog 3 [65].

| Gene | Differential Expression | MBL Subtype | Expressed by | Function |
|------|------------------------|-------------|--------------|----------|
| CD47 | Downregulated | Group 3, Group 4 | CD8+ T cells | Evasion of phagocytosis |
| STAT3 | Upregulated | Not specified | Myeloid-derived suppressor cells | Suppression of pro-inflammatory signalling |
| TGF-β | Upregulated | SHH | Tumour cell | Promotes $T_{Reg}$ cell infiltration, abrogation of NKG2D in NK cells |
| ULBP-4 | Downregulated | Not specified | Tumour cell | NKG2D activating ligand |
| IGF1 | Upregulated | SHH | Tumour-associated microglia | Pro-tumourigenic signalling |
| **ERBB4** | **Upregulated** | **Group 4** | **Tumour-associated microglia** | **Pro-tumourigenic signalling** |
| HLA class I | Upregulated | Not specified | Tumour cell | Antigen recognition by CD8+ T cells, blockade of NKG2DL-mediated NK cell cytotoxicity |
| **PD-L1** | **Upregulated** | **SHH** | **Tumour cell** | Immune checkpoint protein; inhibition of T cell activation |
| IFN-γR2 | Downregulated | Not specified | Tumour cell | Interferon signalling; increased apoptosis and HLA class I expression |
| **CSFR1** | **Upregulated** | **SHH** | **Tumour-associated macrophages** | **Macrophage recruitment and M1 macrophage polarization** |
| CCR2 | Upregulated | SHH | Tumour-associated monocytes and macrophages | CCL2-mediated chemotaxis |
| **IL-22** | **Upregulated** | **Group 4** | **T cells and macrophages** | **B cell activation** |
| B7-H3 | Upregulated | All subgroups | Tumour cell | Immune checkpoint protein; inhibition of T cell activation |

Differential expression is reported with respect to healthy brain tissue, or in **bold text** if measured with respect to other MBL tumour groups.

### 3.1.1. Tumour-Infiltrating Lymphocytes (TILs)

Natural Killer (NK) Cells

NK cells have garnered great interest in recent years as a focal point for understanding tumour immunosurveillance. They have potent tumouricidal properties, capable of mounting rapid responses against cancer cells while simultaneously priming the immune system's adaptive response through the secretion of cytokines such as TNF-α and IFN-γ, which are potent activators of inflammation and macrophage activation, respectively [82]. True to their reputation as a key line of defense against tumour growth, NK cell infiltrates typically comprise a much lower proportion of overall MBL tumour mass in patients with poorer prognoses. Detection of CD56, a biomarker of NK cells, coincides with better survival outcomes alongside with greater mutational burden [83].

Numerous factors contribute to high-risk MBL immune microenvironments' masking effect against NK cell-mediated tumouricidal activity. Natural killer group 2D (NKG2D) receptor signalling is a key player in inducing the cytotoxic arm of the immune system [83]—so it is imperative to note, then, that only a minority of NK cells present in the MBL tumour microenvironment express NKG2D at all. Less than 3% of NK cells present in MBL have detectable surface NKG2D, while peripheral, circulating NK cells almost always do [80,84]. A

drastic downregulation of NKG2D at the tumour site alludes to two possibilities: one, that the reduction of NKG2D expression is a result of NK cell exhaustion, wherein tumour-infiltrating NK cells have long been functionally impaired as a result of chronic overstimulation [85,86], or two—MBLs have successfully co-opted a different molecular mechanism to abrogate NK cell activity.

A common mechanism of immunosuppression exploited by a range of immune-cold TIMEs is through the secretion an excess of Transforming Growth Factor-β (TGF-β); MBL conditions incoming NK cells to tolerate tumour cells despite the presence of surface NKG2D ligands (NKG2DL) [87]. TGF-β appears to influence NK cell behaviour directly: while TGF-β has an extensive number of downstream effects, including mediating T cell regulation and the downregulation of MHC class II receptors [88,89], transduction of a TGF-β dominant negative receptor in NK cells preserves a more tumouricidal phenotype despite exposure to MBL [87]. Although TGF-β exposure does not uniformly deplete *NKG2D* transcript levels in NK cells [90], it appears to induce the production of microRNA-1245. miR-1245 may then go on to attenuate the expression of NKG2D through the degradation of *NKG2D* mRNA or the inhibition of translation initiation [91].

Immunomodulation of NK cell activity by TGF-β only paints a fraction of the entire picture. The abundance of activating NKG2DLs and their cognate receptors with respect to MHC class I expression levels appear to be one of the strongest determinants of the strength of NK cell response against tumour cells. Aberrations in antigen-processing machinery also contribute to producing a highly non-immunogenic environment in MBL. However, while components of the MHC Class I antigen processing pathway are compromised compared to healthy fetal cerebellum tissue [92], measures of activating NKG2DL or HLA class I alone are not predictive of NK cell cytotoxicity. SHH MBL, for example, express a range of NKG2DL without great susceptibility to NK cell-mediated tumour clearance, as long as that HLA class I expression is high relative to often-upregulated NKG2DLs such as UL16 binding protein 2 (ULBP-2) [58].

Taken together, the secretion of TGF-β and expression from HLA class I appear to address the seemingly paradoxical notion that certain NKG2DLs are expressed in MBL—or even overexpressed relative to surrounding noncancerous tissue. The expression of activators of NKG2DLs, such as ULBP-1, ULBP-3, and MHC Class I polypeptide-related sequences A and B (MICA/B) in MBL, are not significantly different from regular brain tissue [80,93]. While they are meant to potentiate inflammatory and cytotoxic capacity, the combined inhibitory effects of both signals are a sufficient safeguard against NK cell-mediated tumour clearance, thereby removing the need to completely eliminate surface expression of relevant activating ligands. A notable exception is MBL's tendency to downregulate the surface expression of ULBP-4, another activator of NKG2DL, on tumour cells. One possibility of why ULBP-4 may be exempt from this trend is its dual function as a γδ-type T cell receptor [94]. γδ T cells, as discussed above, are present and active within the body by the end of the gestational period [95]. They undergo expansion in response to ULBP-4-γδ T cell receptor binding and go on to promote antitumour activity without possible influence from MHC class I-induced cytotoxic downregulation, as these "early" versions of T cells do not rely on peptide recognition to carry out an immune response [96].

As our knowledge of how MBL maintains immune tolerance against NK cells improves, so does our ability to potentially manipulate the tumour microenvironment. Accordingly, NK cells have more recently become an attractive candidate for immunotherapy (NCT02271711) [97].

T and B Lymphocytes

As is the case for NK cells, MBL makes use of various strategies to evade the adaptive immune system. Gururangan et al. [98] highlights the dynamic state of the TIME and its ability to induce a heightened state of immunosuppression under the correct selective pressure. While levels of $T_{Reg}$ cell infiltration exist in MBL tumours compared to noncancer-

ous tissue, they are paired with lower counts of CD4$^+$ T cells overall. Compared to CD8$^+$ cytotoxic T cells, CD4$^+$ T helper cells are much rarer within the tumour microenvironment.

While the effect of TIL populations overall on MBL patient survival is unclear [99], repeated attempts have been made to establish specific lymphocyte subsets as a prognostic marker. A reduction in CD8$^+$ T lymphocytes, for example, often paired with high expression of programmed death ligand 1 (PD-L1), an immune checkpoint molecule, correlates with poorer progression-free and overall survival [100]. These correlative relationships are potentially misleading; while the binding of PD-L1 with its cognate receptor is a well-characterized immune checkpoint that negatively regulates CD8$^+$ T cell-mediated anti-tumour activity [101,102], one study on primary human MBL samples refuted any significance of the PD-1/PD-L1 pathway in MBL, citing an absence of PD-L1 expression by tumour cells and a corresponding minimal infiltration of PD-1$^+$ T lymphocytes present within the TIME [99]. Conversely, further analyses have specified that PD-L1 surface expression in MBL differed depending on molecular characteristics: SHH MBL has the most robust expression of PD-L1, while Group 3 and 4 *MYC*-amplified MBL types have drastically lower expression. IFN-γ also notably upregulates the otherwise minimal expression of PD-L1 of *MYC*-amplified MBL in vivo [59], thereby abrogating T cell-mediated anti-tumour function and eventual apoptosis [103].

It is also pertinent to discuss T lymphocyte recruitment itself, and the features of a given MBL tumour that may inhibit lymphocyte homing onto the TIME. Like previously noted, an immunogenomic analysis by Nabbi et al. [70] postulates that there may be causal relationships between particular oncogenic signalling pathways and the existence of distinct immune clusters among pediatric nervous system tumours. While no direct links have been established at the time of writing in PBTs, some prominent mutations attributed to malignant transformation in MBL have been documented in adult tumours. *Wnt*-activated melanomas, for example, have markedly reduced T cell populations in their TIME compared to their "immune hot" counterparts due to poor dendritic cell (DC)-mediated T cell activation [104,105]. Elsewhere, *MYC*-amplification has been implicated in immune modulation through the upregulation of the CD47 immune checkpoint protein, which can be expressed on tumour cells as a "do not eat me" signal [106–108]. Its binding to its cognate receptor signal-regulated protein-α (SIRPα) impedes phagocytosis and subsequent presentation of tumour-associated antigens on DCs, which is a hard prerequisite to initiating any cytotoxic T cell response [109]. This phenomenon is of possible interest to *MYC*-driven Group 3 MBLs, which have been shown to be vulnerable against CD47 blockades [53]. Should future studies successfully decode what mechanisms exist between a given PBT's driver mutations and their roles in shaping the TIME, it may become a possible avenue to potentiating effector lymphocyte recruitment to the tumour site alongside other immunotherapies or to better understand patient responses to existing treatments.

The relationship between HLA class I proteins and MBL has been discussed in this review in the context of NK cell regulation. While, as also stated in an earlier section, the level of HLA class I protein itself bears no predictive power over the effectiveness of NK-cell mediated anti-tumour activity, its upregulation is associated with poorer outcomes in MBL [110]. At the surface level, this astonishingly opposes the notion that CD8$^+$ T lymphocytes target cells with surface expression of MHC class I protein, and that its subsequent downregulation can be exploited as a strategy for immune evasion by a malignant tumour. Indeed, such a notion is hardly wrong—it, in fact, still appears to hold true for most MBL tumours, which do bear impairments in HLA class I expression and its corresponding antigen processing mechanisms [92]. Meanwhile, Group 3 *MYC*-amplified MBLs are among the most likely to have elevated HLA class I expression, and the apparent pro-tumourigenic effect of this phenotype is enough to confer resistance against CD8$^+$ T cell-mediated cytotoxicity. The same study by Smith et al. [110] deduces that one likely explanation is MHC class I's role in potentiating extracellular signal-regulated kinases (ERK) 1 and 2-related signal transduction, and thus more robust MBL migration and survival that outpaces CD8$^+$ T cell-mediated killing.

Alternatively, the inhibition of NK cell-mediated tumour cell lysis via MHC class I could be what supersedes the effectiveness of CD8[+] T cell activity. While this second explanation is unlikely, as NKG2D expression is notably low in tumour-infiltrating NK cells [80], no study thus far has attempted to specify whether NKG2D expression differs between molecular subgroups, or particularly aggressive forms of MBL, such as the *MYC*-amplified subset. An awareness of the best possible approach towards competing phenotypic effects among immune cell populations will be beneficial, especially in the context of CAR T cells and CAR NK cell therapies.

Lastly, little is known with respect to the role of tumour-infiltrating B lymphocytes in MBL. The proportion of the overall TIME-associated cell population varies; B cells occupy a large component of the TIME in SHH MBL [47]. A separate study, however, reported that B lymphocyte infiltration in Group 4 MBL is comparable to SHH tumours. This may be because the Group 4 TIME comprises a greater cell population overall. The contributions of greater B cell infiltration to apparent higher overall survival rates, decreased CD4[+] T helper type 1 cells, and increased mast cell populations remains unknown [64]. While B cells are not the focal point of more promising immunotherapeutic techniques, demystifying these relationships may prove to be valuable toward treatment design and optimization.

### 3.1.2. Tumour-Associated Macrophages

Because of their pro-tumourigenic role and well-documented association with poor patient outcomes in adult brain tumours, the role of macrophages in the MBL microenvironment has been of particular interest [111]. The presence, prognostic value, and role in tumour progression of tumour-associated macrophages (TAMs) in MBL in particular are one of the most heavily studied—and yet dubiously understood—aspects of the tumour microenvironment. Although MBL-associated macrophages are said to largely emerge from both bone marrow-derived myeloid cells (i.e., BMDMs), while resident microglia generally populate the TIME in lower proportions [112], recent work has alluded to novel and distinct roles for microglia in Group 4 MBL [81] and SHH MBL [57]. Unlike other groups, microglia in Group 4 MBL expresses receptor tyrosine kinase erbB-4 (ERBB4)-activating ligands, whose downstream signalling regulates cell proliferation and differentiation [113–115]. Microglia, therefore, may have an unforeseen role in Group 4 MBL tumourigenesis, whose driver mechanisms remain elusive compared to the better characterized *Shh*- and *Wnt*-driven MBLs. In SHH MBL, tumour-associated microglia occupy a different niche [57], instead driving tumour progression by secreting insulin-like growth factor 1 (IGF1).

Endeavors to describe TAMs and their functions in the context of immunomodulation remains incomplete. Among other molecular groups, SHH MBL has both lower expression of inflammation-related genes such as *CD163* and *CSFR1* alongside the greatest evidence of TAM recruitment [116], and has therefore become the focal point of TAM research in MBL. The detection of colony-stimulating factor 1 receptor (CSF1R) expression in tumours has also garnered modest excitement; CSFR1 signalling has documented tumour-promoting effects [117]. CSFR1 inhibition in adult brain tumours repolarizes its tumour-associated M2 phenotype macrophages toward the "more tumouricidal" M1 phenotype, raising some hope that CSFR1 inhibition may bear synergistic effects with chemotherapy in pediatric SHH tumours [118]. Subsequent animal studies have achieved mixed results: while CSFR1 blockades decrease the degree of TAM infiltration regardless of the model used, therapeutic effects, including the attenuation of SHH MBL tumour progression and promotion of cytotoxic activity at the tumour site, have only been observed in primary tumours [119]. The reduction of TAM infiltrates, meanwhile, has negligible effects on recurring or disseminated tumours [61]; a separate animal model saw no improvement in overall survival, changes in tumour cell proliferation in the leptomeninges, nor production of any synergistic effects when CSF1R inhibition was combined with whole-brain irradiation.

Similar observations were noted while elucidating the role of C-C chemokine receptor type 2 (CCR2) in murine SHH MBL, a specific marker for TAMs of myeloid origin [120]. The genetic deletion of *CcR2* produced a notable absence of macrophages in the tumour en-

vironment without seeing any remarkable differences between overall survival differences, nor a particular synergistic effect with a pre-existing absence of TAMs paired with tumour irradiation [62,121].

TAMs also do not bear a discernible tumourigenic role in Group 3 MBL; in patient-derived xenograft models, while tumour irradiation induced *CSF1* gene transcription, no significant differences in TAM infiltration compared to untreated tumours were seen [61]. Blockade of CSF1R alone or combined with radiation therapy had no effects on survival nor tumour burden. Additionally, while CSFR1 inhibitors have thus far reached phase I/II clinical trials against adult tumours (NCT02526017), drug administration to the pediatric brain also depletes microglia populations in healthy tissue, raising concerns toward possible side effects, including the impairment of CNS development [61].

While the overall MBL TIME has been noted as noninflammatory [122], both M1- and M2-polarized macrophages have been found localized to different areas of the SHH MBL, illustrating a heterogenous microenvironment whose inflammatory status varies depending on intratumoural location [123]. It is, however, worth noting that greater proportions of M1 macrophages are associated with a poorer prognosis in SHH MBL, while tumours with higher M2 macrophage infiltration showed no discernible correlation with survival outcomes [123]. These findings seem to contradict the classical view on macrophage polarization states and their respective roles in the tumour microenvironment: the "classical" M1 phenotype creates a more hostile environment for cancers—in adult brain tumours, a greater M2:M1 ratio is indicative of poorer outcomes [124–126]. M1 macrophages are thought to promote intratumoural cytotoxicity via recruitment and activation of CD8$^+$ T cells and NK cells, while M2 macrophages are more commonly thought as smokescreens against the immune system, secreting immunomodulatory cytokines such as TGF-β and IL-10 [127].

Due to low levels of TAM infiltration in pediatric tumours (as is the case for other components of the pediatric TIME) compared to adult tumours, these arrays of seemingly contradictory observations may be a result of TAM recruitment—regardless of polarization status—being secondary effects of other tumour-driving or anti-tumour processes within the tumour, rather than being the causative factor. M1 macrophages, while bearing tumouricidal properties, may exist in higher quantities in higher-risk SHH MBL not to support tumour growth, but to provide defense against especially aggressive cancers, albeit insufficiently.

Another complicating factor may be other environmental factors related to M1 polarization itself. IFN-γ is frequently implicated in macrophage activation toward the M1 phenotype, inducing reactive oxygen species and TNF-α production, but IFN-γ stimulation also carries oncogenic properties [128,129]. IFN-γ, alongside bearing instrumental importance in both adaptive and innate immune responses, is also implicated in normal CNS—and cerebellar—development via the SHH signalling pathway through transcriptional activation of *Shh* [130–132]. The overactivation of the SHH pathway in the perinatal brain is sufficient to drive tumourigenesis, a murine model of MBL [133]. With this in consideration, it may also be possible that an excess of M1 macrophages is uniquely pro-tumourigenic.

It is also highly likely that a more intensive study of the relationship between TAMs and other components of the tumour microenvironment is required to demystify their role, especially using human-derived in vivo models. While changes in immune cell density distribution following interference with TAM infiltration have been noted, the possible physiological changes have not been closely accounted for. TAM depletion coincides with an increase in CD8$^+$ IFN-γ$^+$ T cells [119], alongside a robust increase in neutrophil populations [62]. While a recent study has reported that high neutrophil-to-lymphocyte ratios are indicative of poor outcomes in both Group 3 and 4 tumours, as is the case for many other tumour types [49,50], little else is known regarding the function of neutrophils in MBL. It is possible that TAMs bear some form of pro- or anti-tumour effect that is "masked" or compensated for by treatments that alter the microenvironment; do the CD8$^+$ cytotoxic T cells or neutrophils, for example, move in to fulfil a similar functional niche

within the TIME? In cancers aside from PBTs, neutrophils induce non-negligible effects that foster either tumour-supportive or tumour-hostile microenvironments [134].

Supposed inconsistencies such as the ones outlined above emphasize the importance of using the existing wealth of literature on adult brain malignancies as a lenient guiding hand, rather than a base framework. Despite not yet being able to identify a clear purpose of tumour-associated macrophages in MBL or clear effects in macrophage-targeted therapies, it would be unsound to wholly disregard TAMs as subjects of interest against MBL. It is clear from our understanding that MBL TIME is still rife with gaps; while many components of the TIME appear attractive as therapeutic candidates, we must tread carefully before making drastic adjustments to our treatment modalities.

### 3.2. High-Grade Pediatric Gliomas

Pediatric high-grade gliomas (pHGGs) are of particular interest for immunotherapy, as some tumours are difficult to excise surgically and few viable therapeutic options have emerged over decades of research [135]. In particular, outcomes for diffuse midline gliomas (DMGs) are quite bleak; both their proximity to the brainstem and tendency to spread into nearby healthy brain tissue render most of these tumours inoperable. Other therapies are rife with their own challenges: despite relying on radiotherapies as our frontline treatment against DMGs, the 2-year survival rate is less than 10% of patients [136].

Our knowledge regarding the pHGG TIME has thus far been limited, but not unexpected: like MBLs with poorer prognoses, pHGGs are typically less immunologically "active" with respect to lower grade gliomas such as pilocytic astrocytomas [137]. While half of all pediatric low-grade gliomas (pLGGs) have significant myeloid cell infiltration and subsequently greater tumour-associated monocyte and granulocyte-mediated inflammation, pHGGs are immune-cold [70]. They bear much lower levels of immune cell infiltrate and have reduced expression of pro-inflammatory markers [138], and are therefore unlikely to be responsive to ICI therapies [139]. Poor responsiveness to ICI therapy by many pHGGs can, in part, be attributed to their weak neoantigen signatures due to low TMBs [140]. However, a subset of a cohort of pHGGs with biallelic deficiencies in mismatch repair genes (bMMRD) are hypermutated. pHGGs with biallelic mutations in the DNA repair genes *MSH6* or *PMS2* can carry over 100 somatic mutations per Mb of genomic DNA. In comparison, the median frequency for all pHGGs hovers closer to 0.1 mutations per Mb of DNA [44]. Consequently, they bear a much higher number of potential targetable neoantigens, and therefore also show better responses to ICI therapies such as PD-1 blockades [141]. bMMRD tumours also tend to have greater myeloid-type immune cell infiltration relative to other pHGG types [70].

In general, pHGGs have limited levels of lymphocyte infiltration compared to adult gliomas of the same tumour grade [142]; lymphocytes make up less than 3% of all CD45$^+$ leukocytes in pHGGs, while adult HGGs may have lymphocytes that comprise up to half of their total CD45$^+$ leukocyte population [63]. However, both pHGGs and pLGGs show hallmarks of greater immunogenicity compared to non-tumour tissue (Table 2), such as increased CD64$^+$ macrophage infiltration and elevated expression of MHC class II [137]; immune-"cold" does not necessarily mean immune-silent.

In this subsection, we will focus primarily on DMGs: they are of particular interest for immunotherapy development given their dismal responses to existing treatments. Approximately 70–80% of DMGs are primarily driven by epigenetic mis-regulation, wherein mutations to the H3 histone—most often a lysine to methionine mutation (K27M)—disrupt the packaging of replicated DNA and its integration into transcriptionally active regions of the genome [143,144]. Because of their highly diffuse, invasive, and pernicious growth that distorts local brain tissue, the available knowledge on the TIME relies heavily on autopsy samples, and more limited data are available on the nature of the DMG TIME at earlier pathophysiological stages and prior to treatment [145].

**Table 2.** Immune signalling components reported in pHGG. ULBP-2/4/5/6, UL16 binding protein 2/4/5/6 [80]; TGF-β, transforming growth factor beta [56]; CCL2, chemokine (C-C motif) ligand 2 [56,63]; IL-β, interleukin 1-beta [56,63]; PD-L1, programmed death ligand 1 [56]; HLA class I, human leukocyte antigen class I [56]; B7-H3, B7 Homolog 3 [56,66].

| Gene | Differential Expression | pHGG Type | Expressed by | Function |
|---|---|---|---|---|
| **ULBP-2/4/5/6** | **Downregulated** | **Not specified** | **Tumour cell** | NKG2D activating ligand |
| TGF-β | Upregulated | Non-DMG | Tumour cell | Promotes $T_{Reg}$ cell infiltration, abrogation of NKG2D in NK cells |
| **CCL2** | **Downregulated** | **Non-DMG, DMG** | **Tumour cell** | **CCR2-mediated chemotaxis** |
| **IL-1β** | **Downregulated** | **DMG and non-DMG** | **Tumour-associated macrophages** | **Pro-inflammatory cytokine signalling** |
| PD-L1 | Upregulated | Non-DMG | Tumour cell | Immune checkpoint protein; inhibition of T cell activation |
| HLA class I | Upregulated | DMG | Tumour cell | Antigen recognition by CD8+ T cells, blockade of NKG2DL-mediated NK cell cytotoxicity |
| **B7-H3** | **Upregulated** | **DMG and non-DMG** | **Tumour cell** | Immune checkpoint protein; inhibition of T cell activation |

Differential expression is reported with respect to healthy brain tissue, or in **bold text** if measured with respect to pLGG expression.

### 3.2.1. Tumour-Infiltrating Lymphocytes

Lymphocyte infiltration in pHGGs is poor. DMGs in particular stand out; the degree of T cell infiltration in these tumours is nearly identical to noncancerous tissue. Lieberman et al. [56] allude to an immunosuppressive mechanism against CD8[+] and CD4[+] T cells; little to no T cell-mediated cell lysis takes place in a CD3/28-activated T cell and DMG coculture model. This is in spite of DMG's surface expression of MHC class I proteins, which is typically an activating ligand for CD8[+] cytotoxic T cells, alongside low expression of the immune checkpoint ligand PD-L1.

NK cells also occupy a relatively minor compartment of the pHGG TIME, and they do not upregulate or downregulate the expression of NKG2D ligands with respect to healthy brain tissue. Certain NKGD2Ls, such as soluble ULBP-2 and cell-surface expression of ULBP-4, however, are significantly upregulated in lower-grade gliomas. The reverse is true in higher tumour grades [80]. A separate study was able to demonstrate effective NK cell-mediated lysis in multiple DMG types despite comparatively low activating NKG2DL expression or surface expression of inhibitory MHC class I vulnerable to NK cell anti-tumour activity—but only at high effector to target ratios that are unseen in vivo [56]. This appears to suggest a possible immune escape mechanism mediated through the expression—or lack thereof—of NKGD2Ls.

One possible route of investigation may be the effects of the H3K27M missense mutation, which produces epigenetic disruptions in part due to the activity of lysine-specific demethylase 1 (LSD1) [143,146,147]. The regulatory activity of LSD1 produces an immunogenic signature in an otherwise immunologically barren midline glioma microenvironment, desensitizing the tumour from immune surveillance. LSD1-mediated histone regulation downregulates the expression of several innate immune ligands, including SLAM family member 7 (SLAMF7), MICB, and ULBP-2 [148]. Indeed, the inhibition of LSD1 markedly improves NK cell-mediated tumouricidal activity [148]. Conversely, although mRNA expression profiles of pHGGs and pLGGs alike appear to be positively correlated with NKG2DL transcript expression, including ULBP-2 and MICB, only pLGGs show discernible increases in protein products of the same genes. Therefore, while it may be tempting to hypothesize that epigenetic modifications mediated by oncogenes such as LSD1 are directly

responsible for NK cell immune evasion, it is highly unlikely that it is the chief explanation for the immune phenotypes we see in pHGGs [80].

What also bears relevance is that the overwhelming majority of infiltrating NK cells lack NKG2D expression entirely, and are therefore unable to interact with the few NKG2D ligands present in the tumour [80]. As discussed with respect to MBL in an earlier section, this may be a sign of NK cell exhaustion [85] or a result of NKG2D downregulation mediated by tumour–NK cell interactions. One potential avenue is elucidating a possible role of TGF-β signalling in mediating NK cell interactions with pHGGs; the overactivation of activin receptor type 1 (ACVR1), a key TGF-β receptor, has been extensively documented in H3K27M DMGs [149–151]. The downregulation of NKG2D on both tumour-infiltrating CD8$^+$ T cells and NK cells has been observed in adult gliomas [152,153]. Some preliminary data suggest it may be possible that a similar effect is taking place in its pediatric counterparts; in vivo DMG secrete comparable amounts of TGF-β.

### 3.2.2. Tumour-Associated Macrophages

Tumour-associated macrophages comprise a notable proportion of the overall DMG immune microenvironment. Unlike MBL-associated macrophages, whose monocyte population largely comprises bone marrow-derived cells from peripheral circulation, bioinformatic analyses suggest that about half of all monocyte infiltration in pHGGs are microglia [47]. They also appear to be markedly less inflammatory than the TAMs that populate adult glioblastoma; relative to their adult counterparts, DMGs have poorer expression of many genes associated with immune cell chemotaxis and the inflammatory response, nor do they express notable levels of proinflammatory cytokines. That said, the TAM population in DMGs does not sport a distinctively pro- or anti-inflammatory phenotype, though there is some measurable CSF1 expression [63]. DMG, in fact, shows little evidence of modulating macrophage function and phenotype at all [56].

DMG cells undoubtedly carry some influence over the monocytes that occupy their tumour microenvironment and may potentially shed some insight toward therapeutic approaches. H3K27M DMGs in particular bear higher monocyte-to-lymphocyte ratios compared to lower grade gliomas, and such a measure may therefore confer some prognostic value. Additionally, DMG patients often have higher systemic monocyte levels in their blood, suggesting that DMGs do induce specific alterations to the immune system, despite its "immunologically barren" status [51].

### 4. Concluding Remarks

Like the tumour itself, the TIME is highly dynamic: it evolves over the course of disease progression, and its composition is prone to change in response to surgical resection or radiotherapies and chemotherapies. Tumour immunotherapy—including the use of immune checkpoint blockades, neoantigen-targeting monoclonal antibodies, and CAR T and NK cell therapies—has garnered great interest over the past decade. A wealth of reviews on the immunotherapy for PBTs in particular have since emerged, among which includes papers by Hwang et al. [154]; Foster et al. [155]; and Wang, Bandopadhayay, and Jenkins [67]. This review intends to act as a complement to these discussions.

What appeared to be overwhelming success in CAR T cell therapy in leukemias [156] highlighted the immense curative potential of immunotherapies against cancer. There is need for improvement, however, as clinical applications of immunotherapy become more widespread in solid cancers. Optimally, immunotherapies would successfully eradicate even aggressive, high-risk, and/or inoperable PBTs while minimizing damage to tissue as delicate and difficult to repair as the structures of the CNS, thereby reducing the incidence of long-term sequelae. Alongside well-examined issues, such as low TMB—and thereby low availability of surface targets—tumour heterogeneity, we must consider how the existing TIME could hinder, or perhaps potentiate the effectiveness of a given immunotherapeutic approach.

Immunotherapies are, by default, founded on an understanding of the TIME, often with respect to effector T lymphocyte function; the clinical benefits of ICI rely on disabling T cell suppression through targeting molecules such as PD-1/PD-L1 or cytotoxic T-lymphocyte-associated protein 4 (CTLA-4)—on the condition that the tumour is, in fact, taking advantage of these regulatory mechanisms to induce immune tolerance [157]. Similarly, we must be highly mindful of what immune cell populations already exist in the TIME and how they may evolve following CAR lymphocyte delivery, as this may impair treatment efficacy or even contribute to resistance [158,159]. Two recent CAR T cell clinical trials for PBTs, described below, illustrate this notion.

One of the first phase I CAR T cell clinical trials (NCT04196413) targeting GD2 disialoganglioside-expressing DMGs successfully curbed disease progression and reduced tumour volume in three out of four patients [21]. While all eventually succumbed to disease, GD2-CAR T therapy prolonged patient survival to 20–26 months following diagnosis compared to an average life expectancy of less than a year for patients receiving standard chemotherapy and radiotherapy [136]. Cytokine panelling and single-cell transcriptomic analyses on patient cerebrospinal fluid (CSF) samples performed over the course of each trial offer novel insights on how the TIME may be conferring treatment resistance or corroborating tumour recurrence. Among all patients enrolled in the trial, CD8$^+$:CD4$^+$ CAR T cell ratios in the CSF generally continued to increase over the course of treatment, suggesting that administered CAR T cell function became impaired over time [160]. The authors also noted the presence of lineage doublets within otherwise distinct T cell and myeloid cell populations in patient CSF; myeloid cells may conceivably be phagocytosing, infused CAR T cells. While such a mechanism of tumour tolerance is thus far undocumented in brain cancers, adult's and children's alike, it has been described as a plausible limitation in immunotherapies against non-small cell lung cancer [161] and colon cancer [162]. One patient in Majzner and colleagues' study showed no significant response to GD2 CAR T cell infusion and had a comparatively immunosuppressive CSF environment, which included both elevated levels of TGF-β and monocyte/microglial cell numbers. Consequently, the authors proposed that the impaired CAR T cell-mediated antitumour effects in this patient may have owed to T cell exhaustion induced by a combination of tonic interferon and PD-1 signalling.

A second phase I trial (NCT04185038) sought to assess the efficacy and safety of B7 Homolog 3 (B7-H3) CAR T cells in DMG patients [163]. Here, multiple reaction monitoring–mass spectrometry performed on patient CSF samples collected over the course of treatment also provided valuable information alluding to TIME-mediated responses to B7-H3 CAR T cell infusion. A number of immunoregulatory peptides were detected in patient CSF following CAR T cell delivery, including programmed cell death ligand 2 (PD-L2), alongside the monocyte/macrophage markers CD14/CD163, or proteins involved in both myeloid and lymphoid cell recruitment, such as a vascular cell adhesion molecule (VCAM-1) and CSF-1.

As more PBT patients are enrolled in immunotherapy trials, both cytokine and bioinformatic analyses will enable robust and accurate predictions of how the tumour and its immune microenvironment will reshape itself in response to immunotherapy, and thereby inform subsequent treatment designs. These approaches stress that our current insights into the PBT TIME remain incomplete and are key to orienting our focus on how—and where—to glean new mechanistic insights into a complex ecosystem. A greater understanding of the tumour's immuno-modulatory strategies, then, strengthens our ability to identify potential interventional targets to modulate ahead of or during immunotherapy [164]. They may even inform development of up-and-coming immunotherapies that can be described as more sophisticated attempts to "hijack" the TIME, such as CAR macrophages, whose conception was borne from the notion that the TIME does not only have to be something to circumvent, but are also a prospective tool we can mould to our advantage [165,166].

**Author Contributions:** Conceptualization, B.Y. and A.D.; writing—B.Y.; writing—review and editing, A.D., H.V. and P.H.S. All authors have read and agreed to the published version of the manuscript.

**Funding:** We acknowledge funding support from the BC Cancer Foundation and from St. Baldrick's Foundation (Empowering Pediatric Immunotherapies for Childhood Cancer-EPICC). A.D. is supported by a Michael Smith Health Research BC Trainee Award (RT-2022-2552).

**Data Availability Statement:** Not applicable.

**Conflicts of Interest:** The authors declare no conflict of interest.

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
