# Peer review of "Pediatric Brain Tumours: Lessons from the Immune Microenvironment"

_curroncol, doi:10.3390/curroncol30050379_

Round 1

Reviewer 1 Report

The paper summarized the components of pediatric brain tumor immune microenvironment and related immunotherapies.

The reviewer has the following questions:

  1. Figure 1 is not mentioned in the main text. What subtypes of pediatric brain tumors are involved in Figure 1A? What’s the control for upregulation and downregulation?
  2. In Figure 1B, is the morphology of low-grade and high-grade tumors similar? I suggest labeling lymphocytes and showing lymphopenia by drawing different proportion.

3.       Would another figure for the immunotherapies to medulloblastoma and high-grade gliomas help connect points in this review paper and improve the clarity?

  1. Why did the authors introduce new information about CAR-T in conclusion remarks rather than in discussions?
  2. For the paragraph: “In the context of PBTs themselves, it is also noteworthy that the pediatric BBB does not share the same qualities as its adult counterpart; the younger, functional BBB readily allows plasma-derived proteins regulated, transcytosis-mediated access to the brain parenchyma, but the aged, “leaky” BBB is more permissive to the entry of potentially neurotoxic proteins from peripheral blood [12].”

It might be worthwhile to discuss what developmental/age-related changes make BBB leaky in adults.

The authors’ description of establishing the prognostic value of any molecular/cellular aspect of PBT seems convoluted. The particular discussion can be simplified separately for better comprehension.

Readers might be interested to learn about any well-known instance of immunotherapy conferring clinical benefit over chemo/radiotherapy in terms of survival in patients with PBT.

Reviewer 2 Report

Yao et. al have done a great job discussing the immune micro-environment in pediatric brain tumors with a focus on Medulloblastoma and High-grade gliomas. The authors also discussed how immune-tolerance is developed in pediatric brain tumors and summarized the current challenges in immunotherapy. The authors also briefly discussed the therapeutic implications of the tumor immune-microenvironment, describing results from few ongoing trials of CAR-T therapy, checkpoint blockade etc. Overall, it is a well -written, informative article.

Reviewer 3 Report

The manuscript "Pediatric Brain Tumours: Lessons from the Immune Microenvironment" by Betty Yao and colleagues contains an exhaustive, timely and well written review of the current knowledge, techniques and problems related to immunotherapy of Pediatric Brain Tumours. Despite the text is well written and clear, the large mass of information may sometimes overwhelm the general reader. The addition of figures summarizing the information provided for each of the immune cells and tumours considered will be most desirable.

Minor point

row 520 " a subset of for a cohort" please correct " a subset of a cohort".
